# Brain activity in Cluster N and the hippocampus in non-migratory zebra finches completing a spatial orientation task using magnetic compass information

Madeleine I.R. Brodbeck[1]*, Rachel Muheim[2], Atticus Pinzon-Rodriguez[2], Scott A. MacDougall-Shackleton[1]

**1** Department of Psychology, Advanced Facility for Avian Research, University of Western Ontario, London, Canada, **2** Department of Biology, Lund University, Lund, Sweden

* mbrodbec@uwo.ca

## Abstract

The songbird brain region called Cluster N has been implicated as important for low-light vision and/or the perception of Earth's magnetic field for migratory orientation. This region of the visual Wulst is active in migratory songbirds under dim light conditions and an intact Cluster N is necessary for magnetic compass orientation in a nocturnally migrating songbird species. Given that magnetic field information is useful for orientation outside of a migration context, it is likely that Cluster N processes magnetic compass information more generally. Zebra finches (*Taeniopygia guttata*) can use magnetic compass cues to find food in a plus maze, even though they are not nocturnal migrants. Our objective was to determine if Cluster N is active when zebra finches use their magnetic compass to orient in a plus maze. Zebra finches were tested under three conditions: i) a static magnetic field that reliably indicated the food location, ii) a sweeping magnetic field, or iii) a vertical magnetic field. The latter two conditions did not provide any directional information. Brains were collected following the task and processed to label the immediate early gene *zenk*. We found elevated Zenk immunoreactivity in the forebrain region defined as Cluster N in other species and in the hippocampus. We found no differences in Zenk between the three magnetic field conditions, providing no conclusive evidence for whether Cluster N is involved in the processing of magnetic compass information. In conclusion, our results are consistent with the idea that Cluster N is not a brain area restricted to nocturnally migrating songbirds, but that it is also found in non-migratory birds carrying out a spatial orientation task, possibly involving magnetic compass cues, under dim light conditions.

**Data availability statement:** Data for this project are available on OSF at: https://osf.io/y7uxw/?view_only=d6a0059d7c044e1ab74ba-44b253a7892.

**Funding:** This research was supported by Natural Sciences and Engineering and Research Council of Canada (https://www.nserc-crsng.gc.ca/index_eng.asp) Discovery Grants awarded to SAM-S 21738, and by grant 2019-03620 from the Swedish Research Council (Vetenskapsrådet; https://www.vr.se/english.html) to RM. MIRB was supported by a Natural Sciences and Engineering Research Council of Canada Doctoral Scholarship.

**Competing interests:** The authors have declared that no competing interests exist.

## Introduction

Research on the brain area(s) involved in the perception and processing of magnetic compass information in migratory songbirds has implicated that the brain region called Cluster N is important for low-light vision and/or the perception of Earth's magnetic field [1–9]. This region of the visual Wulst receives input from the retina, the putative site of the magnetoreceptors of the avian magnetic compass [10–12], via the thalamofugal visual pathway [13]. Cluster N activation has been suggested to be present primarily in nocturnally migrating songbirds, exhibiting migratory restlessness [8–9], during the migration season in low light conditions [1–9]. However, there are notable observations suggesting that Cluster N activation is not restricted to migratory birds exhibiting migratory restlessness, but that it occurs more widely among songbirds exposed to dim light conditions. In many studies, Cluster N was activated in birds sitting still, but awake, in a dimly lit cage, whether it was during the migration season, during the pre-migratory season before the onset of migration, or during the non-migratory season [1,2,5,6]. Moreover, elevated activation of Cluster N has also been observed in non-migratory birds, like Sardinian warblers (*Curruca melanocephala*; 2) and zebra finches [(*Taeniopygia guttata*; 14)]. These observations contradicted the initial conclusion that Cluster N is specific to nocturnally migrating songbirds [1] that was based on initial failure to find Cluster N activation in zebra finches and canaries (*Serinus canaria*). Cluster N activation thus seems to be much more common and by far not restricted to nocturnally migrating songbirds.

Direct evidence for an involvement of Cluster N in processing magnetic compass information is limited to one study showing that lesions to Cluster N disrupt the use of the magnetic compass during orientation [15]. European robins (*Erithacus rubecula*) with chemical lesions to Cluster N were unable to correctly orient when tested for magnetic compass orientation [15]. However, the same birds were still able to orient using their star compass, which strongly suggests that Cluster N is critical for processing geomagnetic cues for orientation [1]. Yet, comparisons of neuronal activation in Cluster N between birds exposed to different magnetic field conditions, like horizontally sweeping magnetic fields, zero magnetic fields, and the natural geomagnetic field, have not revealed any or only marginal differences in activation patterns in Cluster N between conditions [2,14].

Despite the focus on orientation during migration, magnetic field information is potentially useful in other contexts and may be used by birds and other animals for orientation outside of a migration context to locate useful aspects of their environment [(e.g., food sources, nest places; [16–18])]. Magnetic field perception is not limited to migratory bird species, but is also present in non-migratory birds like chickens [(*Gallus gallus*: 19)], homing pigeons [(*Columbia livia domestica*: 20)], and zebra finches [18,21,22]. It is therefore possible that Cluster N processes magnetic cues more generally. Prior investigations of Cluster N activation in non-migratory birds lacked stimuli involving active magnetic compass use. For instance, the initial study by Mouritsen et al. [1] merely observed zebra finches and canaries in an arena prior to euthanasia. The absence of immediate early gene expression in the visual Wulst region (where Cluster N is found) might reflect the lack of magnetic compass use while a bird is stationary.

Following this evidence, a key question remains: does Cluster N exhibit activation in non-migratory birds actively using their magnetic compass, or is it exclusive to nocturnally migrating songbirds? Further, if Cluster N is not activated in non-migratory birds during magnetic compass orientation, which brain regions are instead used to process magnetic compass information? To answer these questions, we used zebra finches trained to relocate a hidden food reward in a plus maze using directional magnetic compass information. Zebra finches are songbirds that are not nocturnal migrants and therefore provide a compelling comparison to nocturnally migrating songbirds that exhibit Cluster N activity at night. There is convincing evidence that zebra finches have a light-dependent magnetic compass just like nocturnally migrating songbirds [18,21–23]. Zebra finches trained to locate a food reward using magnetic compass information in a spatial orientation task were well oriented along the trained magnetic compass axis when tested under low-irradiance green, but not red, light, while the presence of a 1.4 MHz radio-frequency electromagnetic (RF)-field disrupted their orientation [18,22,24]. This orientation disruption provided strong evidence for the involvement of radical-pair reactions in the primary magnetoreception process in zebra finches and aligns well with the patterns observed in nocturnally migrating birds [18,21–24]. Findings of expression patterns of cryptochromes in the zebra finch retina [25] further support the notion that cryptochromes act as the primary magnetoreceptor, just like in nocturnally migrating songbirds. In essence, this work suggests that the magnetic compass responses of zebra finches mirror those of migratory birds and lends strong support to the idea that light-dependent, radical-pair-mediated magnetoreception is a characteristic shared by a variety of bird species, including those that are not migratory.

Given the shared sensory properties of the magnetic compass of zebra finches and migratory songbirds, it is plausible that the neuronal processing of this information occurs in similar structures in the brain of zebra finches and nocturnally migration songbirds. Keary & Bischof [14] demonstrated that zebra finches exhibit neural responses to magnetic field cues. They exposed birds to either a static or sweeping magnetic field with the local intensity and inclination of Earth's magnetic field for an hour under each condition. To analyze potential differences in brain regions, including the hippocampus and the hyperpallium densocellular (HD), they used the immediate early gene *c-fos* as an activity marker. The sweeping magnetic field led to enhanced activation in a rostral segment of the dorsomedial hippocampus and a rostral portion of the HD [14]. These findings align with previous research that highlighted the role of sensory systems in perceiving the Earth's magnetic field. Moreover, Keary & Bischof [14] proposed that these sensory mechanisms, though perhaps less intricate, might also be present in non-migratory songbirds. In essence, their work implies a potential shared mechanism between both migratory and non-migratory birds.

In the current study, we extended this work by examining the activation of the immediate early gene *zenk* in the brains of zebra finches using magnetic compass information to solve a spatial orientation task. We tested birds in different magnetic field alignments of a static magnetic field to ensure that the birds used their magnetic compass for orientation and compared them to birds tested in a vertical magnetic field (no directional information) and birds exposed to a sweeping magnetic field (no consistent directional information). We then examined activation of the immediate early gene *zenk* in the brains of these birds to assess two possible outcomes. First, zebra finches might show expression in Cluster N while actively perceiving magnetic field information, demonstrating that Cluster N is not unique to nocturnally migrating birds. Second, zebra finches might not show expression in Cluster N, which would strengthen the argument that Cluster N is important for night vision and magnetic orientation only during migration. We also investigated Zenk expression in three subregions of the hippocampus, which is known to be critical for spatial cognition and could be a complimentary or alternate area for magnetic compass information to be processed. The hippocampus has been implicated to be important in zebra finches during spatial orientation [(see 26)], and it has been suggested that information between the hippocampus and Cluster N are integrated [27].

## Methods

### Subjects

We used 23 adult male zebra finches (>6 months of age) from a captive breeding colony. Birds were housed in an outdoor aviary at Stensoffa field station near Lund, Sweden. For the experiments, birds were brought inside a wooden hut with

access to natural light conditions and housed in groups of 3−6 birds in holding cages (0.8 x 0.48 x 0.83 m). All experiments were carried out with approval from Malmö-Lund Ethical Committee (permit nr. M 24−16) and conducted in accordance with Swedish legislation and the ARRIVE guidelines.

## Experimental Set up

Birds were trained and tested in a plus maze made of a wood frame and netting with dimensions 1.2 x 1.2 m (see 18 & 22 for detailed methods). Each arm of the plus maze contained a red food cup, where a food reward (millet) could be hidden. The surroundings of the maze were covered by a white sheet of fabric to eliminate the possibility of birds using any extra-maze visual cues. All birds completed the task in the same low-light intensity (27 mW/m$^2$) green light (521 nm) as previous work has shown zebra finches successfully orientating under this light [18,23]. This light intensity is about one order of magnitude (10–20 x) higher than the dim light conditions used in other studies, but considerably lower than bright indoor light conditions (see S1 Table).

The maze was centered in a magnetic coil (2 x 2 x 2 m) with a Merritt design. We used three magnetic field conditions: static magnetic field, sweeping magnetic field, and vertical magnetic field. The static magnetic field was set to have the same intensity and inclination as Earth's magnetic field (intensity ~50.5 µT; inclination 69.8°) and could be directed towards any of the maze arms, corresponding to different geographic directions (North, South, East, West). The sweeping magnetic field condition consisted of a magnetic field of the same properties (inclination and intensity) as the natural magnetic field but was horizontally sweeping from South to East to North and back to East and South in 36 steps of 10° at 0.083 s per step (3 s for one sweep of 360°). In the vertical magnetic field, the horizontal component of the magnetic field was cancelled, leaving only the vertical component pointing downwards, providing no directional magnetic compass information as opposed to pointing towards a particular arm of the maze in the static magnetic field.

## Behavioural procedure

We first trained birds to relocate a food reward under the static magnetic field condition. Birds naïve to the training apparatus were first accustomed to the maze in the absence of any directional information from the magnetic field. Birds were placed in one arm with access to food until they learned to search for food in the red food trays. To encourage the birds to search for food, access to food was restricted in their home cage the night before acclimation or training started. Once a bird had learned the task, we trained birds to relocate a food reward under the static magnetic field condition (see supplemental information in 22 for detailed methods). Birds were placed into a release device in the middle of the maze, in complete darkness, to allow time for the experimenter to leave the room and observe the bird completing the task with an overhead camera. The lights were turned on, and after 30 s the bird was released from the release device. Birds searched the arena until they located the food reward. If a bird entered the correct arm of the plus maze, they were rewarded with 10–15 s of food access. If a bird entered an incorrect arm of the plus maze without a food reward, lights were turned off for 3–5 s. Training trials were considered successful if the bird entered the correct arm of the plus maze within 3 minutes, and without entering more than 8–10 arms in total. Birds learned the task rapidly, in 3–4 training trials, at about 90 s per trial. Between training trials, the location of the correct arm, the food reward, and the alignment of the magnetic field were shifted 90° clockwise or counterclockwise to eliminate the possibility of finches using any other cue than the alignment of the magnetic field to solve the task. Individual birds were trained to find the food reward in either one of four magnetic field direction (magnetic North, mN, mS, mE, or mW; see supplemental information in 22 for detailed methods). The birds learn the task in only 2–4 training trials, thus learning acquisition with this training procedure is very rapid. Repeated training does not lead to better performance, which is in contrast to traditional conditioning assays where the learning curves of individuals are important measures.

Following successful training, birds were tested in a probe trial on the following morning under one of the three experimental magnetic field conditions described above. The group of birds tested in the static magnetic field experienced the

same conditions as in training, with magnetic North directed towards either geographic North, East, South or West, and was expected to be able to use the magnetic compass for orientation. Birds tested in the sweeping magnetic field and vertical magnetic field groups were not expected to be able to orient, as they could not gain any useful directional information from the magnetic field to complete the task, while still being exposed to magnetic field information. During the probe trial, the birds were allowed to search the maze without a food reward present for 30 min to allow for the transcription and translation of immediate early genes in the brain regions of interest. We recorded the behaviour of the birds with a video camera above the maze, and later analyzed the activity and orientation by video tracking.

## Brain preparation and immunohistochemistry

Immediately after the end of the probe trial, the birds were transported to another room in a black, light-tight cloth bag and placed in a dark cage for another 30 min before brain collection. Birds were rapidly decapitated, and the brains were removed from the skull and placed in a vial containing 4% paraformaldehyde for 24 h. Then, the brain was transferred to 30% sucrose for cryoprotection. Once a brain was fully saturated and sunk (24–48 h), it was frozen on pulverized dry ice and stored in a −80 °C freezer. Brains were then shipped from Lund to Western University in a liquid nitrogen cryoshipper, where they were then stored at −80 °C until processing.

All brains were sliced at a 40 μm thickness in the coronal plane at −20 °C using a cryostat. Brain sections were collected in 24-well tissue culture trays containing 0.1 M phosphate buffered saline (PBS). Brain sections were collected beginning at ~5 mm anterior to the cerebellum and ended when the cerebellum was clearly visible. We collected every brain section in two series: one to process with immunohistochemistry to label Zenk, and the second to process with immunohistochemistry to label for GluR-1. GluR-1 clearly labels the dorsal nucleus of the hyperpallium (DNH), a structure useful for locating Cluster N [1,2,4,5,7–9,27].

Immunohistochemistry was completed in runs of one to two brains at a time, in a pseudorandom order balanced across the three groups to remove processing order as a confounding factor in staining intensity. All brain processing was conducted while blind to treatment group. We washed free-floating sections two times, 5 min per wash, in 0.1 M PBS. Sections were then incubated for 15 min at room temperature with agitation in 0.5% $H_2O_2$ to eliminate endogenous peroxidase. Sections were washed again three times (5 min each) in 0.1 M PBS and then incubated for 1 h at room temperature with agitation in 10% normal goat serum diluted in 0.3% Triton-X in PBS (PBS-T). Sections were incubated at room temperature for about 24 h with either anti-Zenk (kindly provided by David Keays [28]; diluted 1:10,000) or anti- GluR-1 (RRID: AB_2113602, AB1504, EMD Millipore Corp, diluted 1:250 in 0.3% PBS-T). Sections were washed three times and then incubated for 1 h at room temperature with agitation in biotinylated goat anti-rabbit secondary antibody (BA-1000, Vector, diluted 1:250 in 0.3% PBS-T). Sections were again washed three time and then incubated for 1 h at room temperature with agitation in avidin-biotin horseradish- peroxidase complex (Vectastain ABC, Elite Kit, diluted 1:200). Sections were washed a final two times and then visualized with the glucose-oxidase-DAB (3,3'-diaminobenzidine) method. Sections were exposed to the DAB reaction for 5 min before being rinsed four times and float-mounted on Superfrost Plus Excel slides (Fisherbrand) and left to dry at room temperature for approximately 24 h. Sections were dehydrated in a sequence of increasing ethanol concentrations and then cleared in a xylene substitute for about 24 h before being cover-slipped.

## Cluster N Visualization and Cell Counting

All photomicrographs and cell counts were collected blind to the group of the birds. We used GluR-1 staining to facilitate locating the DNH, a structure with intense GluR-1 staining located within Cluster N [1,2,4,5,7–9,27]. Here, we only collected coronal sections to facilitate analysis of the hippocampus, which is easily damaged in sagittal sections. To choose where to sample Cluster N in coronal sections, we processed an additional zebra finch brain not part of this experiment. We collected every section of this additional brain, and processed half of the sections with GluR-1 (with the same

procedure as described above), and Nissl-stained the other half with thionin. We used these sections as a reference to carefully select the sampling windows, or fields of view (255 μm by 190 μm), for visualizing Cluster N in coronal sections of the experimental brains. This reference brain also allowed us to locate another set of landmarks for a more anterior sampling area in coronal sections, compared to the more posterior location of the DNH. In the more anterior sections, we chose two sampling locations or fields of view where the song-control region Area X was the largest in GluR-1-stained sections (Fig 1). In the more posterior sections, we only chose one sampling location or field of where the DNH was the darkest (Fig 1). Thus, in Cluster N, we had three subregions that we named anterior ventral, anterior dorsal, and posterior.

Photomicrographs in the anterior ventral and anterior dorsal subregions of Cluster N were taken from three sequential brain sections, from ~4.4 mm to ~4.2 mm anterior to the cerebellum in each hemisphere. Photomicrographs in the posterior subregion of Cluster N were taken from three sequential brain sections, from ~3.5 mm to ~3.2 mm anterior to the cerebellum in each hemisphere. Thus, a total of 18 photomicrographs (three subregions, three brain sections per subregion, two hemispheres) were taken per bird in Cluster N.

Counts of Zenk-labelled cells were performed using an automated macro instruction written for ImageJ [29]. Parameters for labelled cell size were calculated by obtaining measures from a random set of 60 photomicrographs. The parameters for a labelled cell to be counted based on size were set to one standard deviation above and below the mean of this

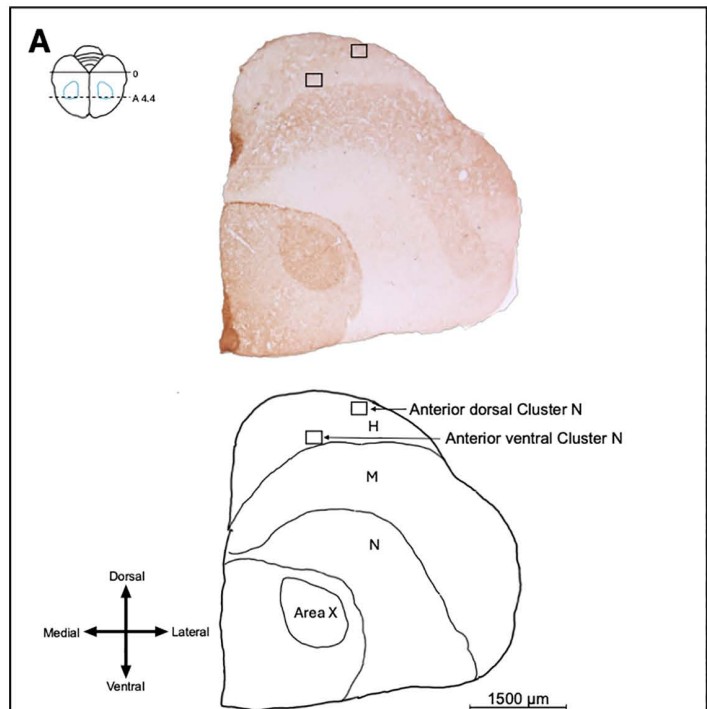
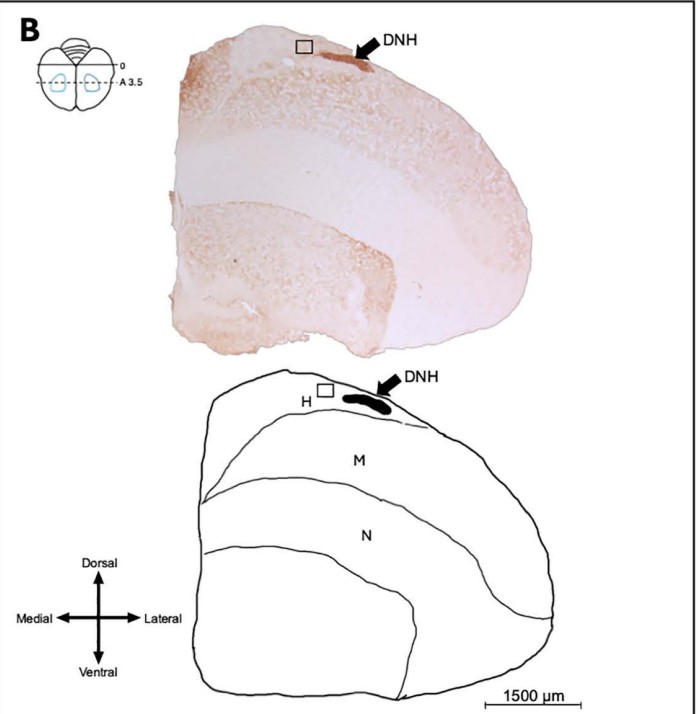

**Fig 1. Photomicrographs of brains stained for GluR-1 and line drawing of brains indicating where pictures of Cluster N were taken.** Two pictures were taken at a more anterior location of Cluster N (anterior dorsal; anterior ventral), and one picture was taken at a more posterior location in Cluster N. Top right schematic in both panels shows an overhead view of brain where sections were collected. Blue outlines show the estimated position of Cluster N. Bottom left symbol in both panels is an anatomical orientation compass showing dorsal (top), ventral (bottom), medial (left), and lateral (right). Panel A: Photomicrograph and line drawing at anterior subregion of Cluster N. Rectangles show the fields of view for anterior Cluster N (255 μm by 190 μm). Photomicrographs were chosen in this series where Area X appeared the largest. Panel B: Photomicrograph and line drawing at posterior subregion of Cluster N. Rectangles show the fields of view for posterior Cluster N. Photomicrographs were chosen in this series where DNH appeared the darkest. The black arrow points to the DNH, a structure useful in locating Cluster N. H = hyperpallium, M = mesopallium, N = nidopallium, DNH = dorsolateral nucleus of the hyperpallium.

subset of cells (mean = 23.80 µm²; SD = 7.70 µm²). Thus, cell size parameters were set to 16.10–31.50 µm². Parameters for circularity were 0.4–1.00. All labelled cells fitting these parameters in each field of view were counted. For each bird, we calculated the average number of cells per field of view and used the overall average per subregion, per bird for statistical comparisons between groups.

## Hippocampus Visualization and Cell Counting

Photomicrographs were taken in three subregions of the hippocampus: the dorsolateral hippocampus (dl), dorsomedial hippocampus (dm) and the ventral hippocampus (v) (see Fig 2). Six photomicrographs were taken in each subregion of the hippocampus. Three of the photomicrographs were taken at a more anterior location (~2.5–2.3 mm anterior to cerebellum), and the other three of the photomicrographs were taken at a more posterior region (~1.5–1.3 mm posterior to the cerebellum). Thus, a total of 18 photomicrographs were taken per bird in the hippocampus.

Counts of Zenk-labelled cells were carried out as in Cluster N. Cells from 60 random photomicrographs were used to determine the parameters for cell size for each subregion. The parameters for a labelled cell to be counted based on size were set to two standard deviations above and below the mean of this subset of cells for each subregion (dlHp mean = 26.28 µm²; SD = 9.76 µm²; dmHP mean = 49.26 µm², SD = 20.36 µm²; vHp mean = 47.34, SD µm² = 17.96

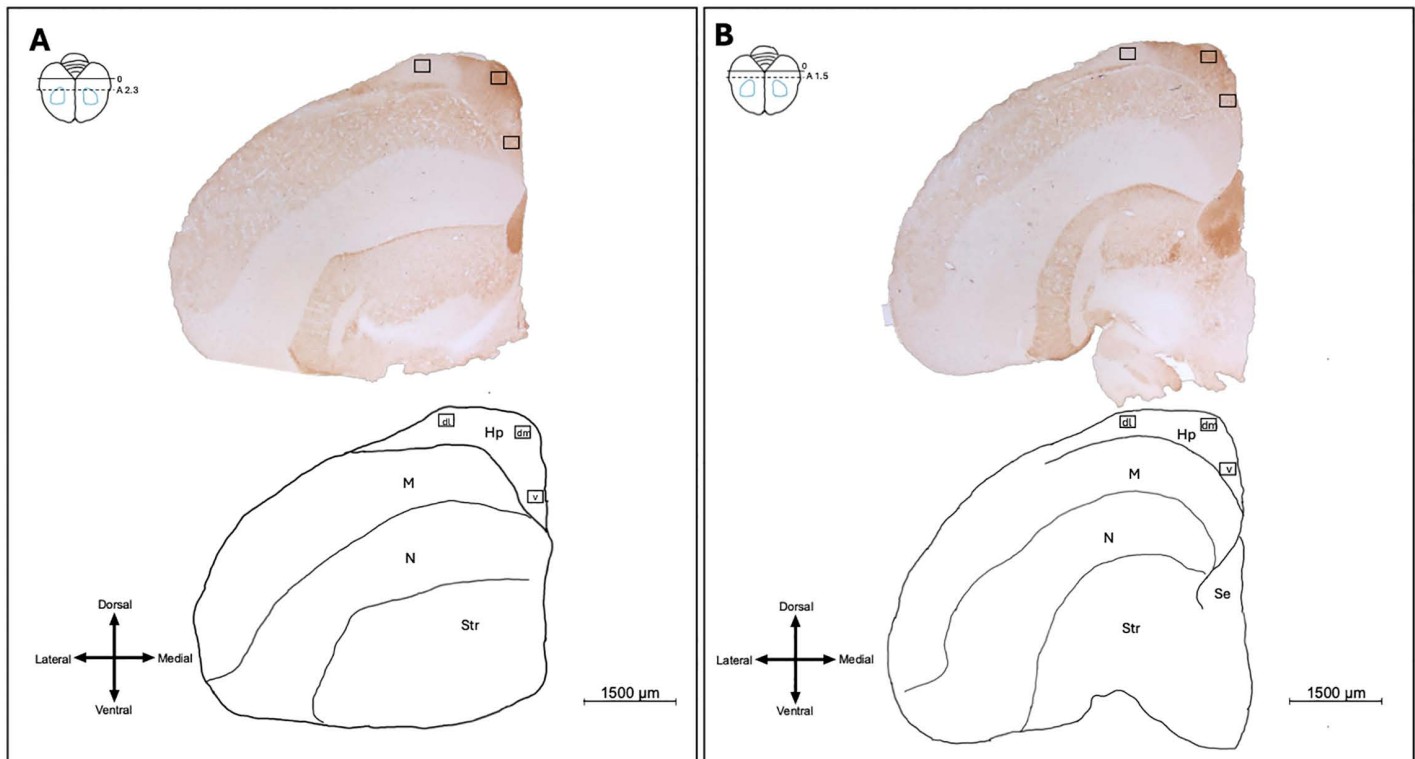

**Fig 2. Photomicrographs of brains stained for GluR-1 and line drawing of brains indicating where pictures of the hippocampus were taken.**
Three pictures were taken at a more anterior location of the hippocampus, and three picture was taken at a more posterior location of the hippocampus. Top right schematic in both panels shows an overhead view of brain where sections were collected. Blue outlines show the estimated position of Cluster N. Bottom left symbol in both panels is an anatomical orientation compass showing dorsal (top), ventral (bottom), lateral (left), and medial (right). Panel A: Photomicrograph and line drawing at anterior subregion of the hippocampus. Rectangles show the fields of view for anterior hippocampus (255 µm by 190 µm). Panel B: Photomicrograph and line drawing at posterior subregion of the hippocampus. Rectangles show the fields of view for posterior hippocampus. Hp = hippocampus, dl = dorsolateral, dm = dorsomedial, v = ventral, M = mesopallium, N = nidopallium, Str = striatum, Se = septum.

μm$^2$). Thus, cell size parameters were 11.42–83.27 μm$^2$ for vHP, 8.53–89.99 μm$^2$ for dmHp, and 6.77–45.80 μm$^2$ for dlHp. Circularity limits were set to 0.4–1.00. All labelled cells fitting these parameters in each photomicrograph (field of view) were counted. For each bird, we calculated the average number of cells per field of view per subregion and used the overall average per bird for statistical comparisons between groups.

## Statistical Analyses

**Behaviour.** The orientation of each bird relative to geographic North in the maze was calculated from the time (number of frames) it spent in each of the four maze arms during the first 90 s of the probe trial, as described in Muheim et al. [22]. For the static magnetic field group, the resulting directions were then recalculated relative to magnetic North (mN = 0°, taking into consideration that different individuals were tested in different magnetic field alignments), and relative to the trained magnetic compass direction (correcting for whether the bird was trained to mN, mE, mS, or mW). These conversions were not possible for the sweeping and vertical magnetic field groups, since the magnetic field had no fixed reference in these conditions. To assess whether locomotor activity differed between magnetic field treatment groups during the orientation task, we quantified total movement for each bird during the 30-min maze exposure. Activity was measured as the total activity of the bird over the 30-minute period, measured in pixels. Activity levels were compared across treatment groups using a Kruskal–Wallis one-way analysis of variance, as the data did not meet assumptions of normality.

For each experimental condition, the mean orientation of the group of birds was calculated using vector addition from the individual mean directions. For all groups, it was determined whether a unimodal or bimodal distribution best fitted the orientation data by calculating the mean vector length for the two distributions. For the distribution that best described the data, i.e., the distribution with the longer mean vector, the Rayleigh test was performed to test for significance [30].

**Zenk Immunoreactivity.** Tests of significance were performed at α = 0.05 using Graphpad Prism version 10. Several statistical analyses were conducted to investigate different aspects of the data. First, paired t-tests were employed to assess left-right hemispheric differences within each subregion (field of view within brain subregions). One bird had missing values due to tissue processing in the posterior dorsal cluster N subregion, thus, to examine group differences in Cluster N we used a linear mixed effects model rather than ANOVA. This analysis incorporated subregion (anterior dorsal, anterior ventral, and posterior dorsal) and experimental group (magnetic field, sweeping field, vertical field) as fixed effects, while subject served as a random effect to control for repeated measures. Additional investigations into potential differences between the experimental groups were carried out using three separate one-way ANOVAs for each subregion of Cluster N. Additionally, in the static magnetic field group a correlation between degrees oriented away from the trained orientation (i.e., trained orientation subtracted by observed orientation), and cell counts in Cluster N was run. To test whether the orientation performance of the static magnetic field group correlated to Zenk expression in Cluster N, we first calculated the deviation from the rewarded axis of orientation. Next, we analyzed the correlation between this deviation and cell counts in three areas of Cluster N (dorsal anterior, ventral anterior, and posterior) using a linear correlation (Pearson's). For the analysis of cell counts in the hippocampus, three two-way ANOVAs were conducted per subregion. These analyses used experimental group and neuroanatomical orientation (anterior vs. posterior) as factors.

## Results

### Behaviour

The zebra finches tested in the static magnetic field oriented axially relative to the trained magnetic compass direction, which, however, was not statistically significant (α = 172°/352°, r = 0.553, p = 0.06; Fig 3). Birds tested in the vertical magnetic field were not significantly oriented in any topographic direction in the maze (α = 25°, r = 0.406, p = 0.3), and birds tested in the sweeping magnetic field did not significantly orient towards topographic North in the maze (α = 25°, r = 0.602, p = 0.08). Birds tested in the static magnetic field similarly did not show a significant orientation relative to North in the

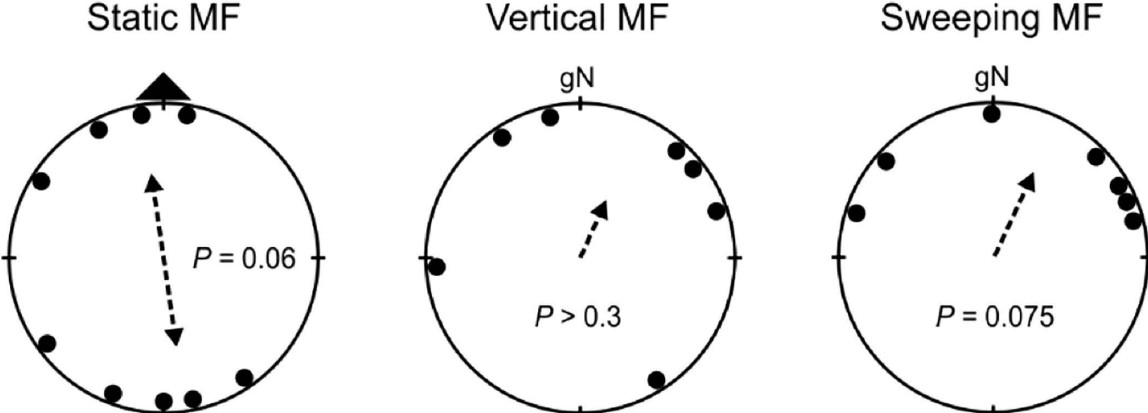

**Fig 3. Orientation of zebra finches trained to magnetic compass cues in a 4-arm maze.** Each dot represents the orientation of an individual bird (see behavioural analysis section), either relative to the trained magnetic compass direction (static magnetic field (MF) group) or relative to geographic North (vertical and sweeping MF groups). The arrow in the middle of each circular plot denotes the mean direction of the birds in each experimental group. The length of the arrows is proportional to the mean vector length (radius of circle = 1). Single-headed arrows indicate unimodally distributed groups, double-headed arrows indicate bimodally distributed groups. None of the groups was significantly oriented according to the Rayleigh test (P-values ≥ 0.05).

maze (α = 136°/316°, r = 0.201, p = 0.70; not shown). Activity levels differed significantly between magnetic field treatment groups (Kruskal–Wallis test, p = 0.04), with birds tested in the natural magnetic field showing the highest levels of activity and birds tested in the vertical magnetic field showing the lowest levels (Fig 4).

### Cluster N

Zebra finches in all groups showed substantial Zenk expression in the region of the Wulst defined as Cluster N in other species (see Fig 5, Fig 6). Two of the brains were excluded from analysis of interhemispheric differences as it was inconclusive which hemisphere was which during histological processing. An additional bird was excluded in the posterior dorsal subregion as the tissue was damaged during histological processing, and no cell counts were obtained for that particular subregion in that particular bird. Paired t-tests showed no interhemispheric differences in any of the subregions (dorsal anterior Cluster N (Table 1), thus scores for each hemisphere were averaged together, and the scores from the two earlier excluded brains were also included in subsequent analyses.

No significant difference in Zenk immunoreactivity between experimental groups was observed in the mixed effects analysis (Fig 6; F(2,20) = 0.18; p = 0.84). A significant difference was observed by brain subregion (F(1.48, 28.89) = 49.01; p < 0.0001). No significant interaction between experimental group and brain subregion was observed F(4, 39) = 1.28; p = 0.29).

In the linear correlation between the orientation performance of the static magnetic field group and Zenk expression in Cluster N, there was no significant correlation in any of the areas in Cluster N: Dorsal anterior: $r^2 = 0.0003$, p = 0.96; Ventral anterior: $r^2 = 0.28$, p = 0.14; Posterior: $r^2 = 0.007$, p = 0.83.

### Hippocampus

Two of the brains were excluded from interhemispheric differences as it was inconclusive which hemisphere was which during histological processing. Paired t-tests showed no interhemispheric differences in any of the subregions (Table 2), thus scores for each hemisphere were averaged together, and the scores from the two earlier excluded brains were also included in subsequent analyses.

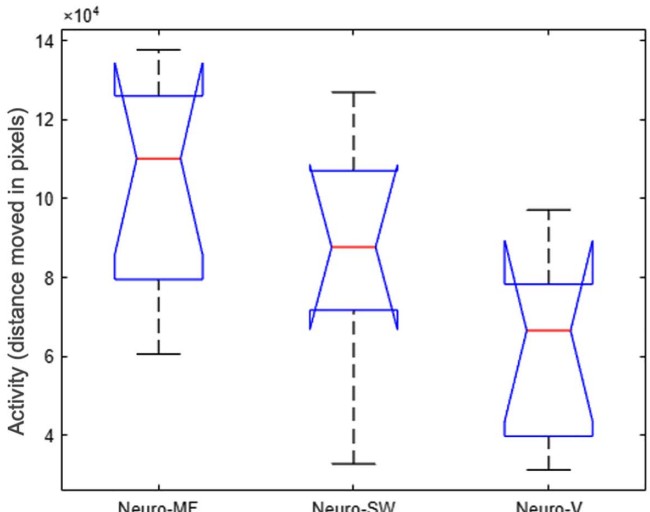

**Fig 4. Activity differences between magnetic field conditions.** Activity levels of zebra finches across magnetic field treatment groups during the 30-minute maze orientation task. Activity here is defined as the total activity of the bird in the 30 minute period, measured in pixels. Birds in the magnetic field group moved significantly more than birds in the other two groups, as reported by a Kruskal-Wallis ANOVA.

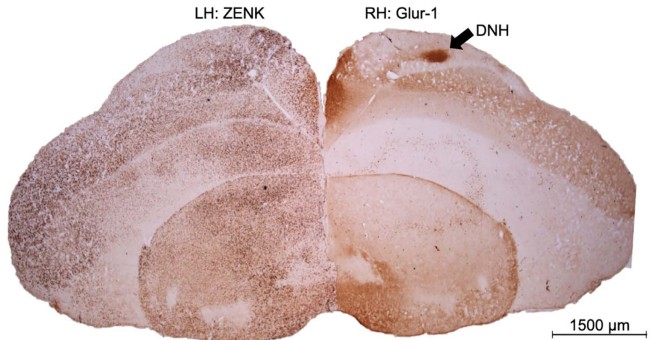

**Fig 5. Representative photomicrograph of a Zebra Finch brain from the experiment.** The left hemisphere was stained for Zenk, and the right hemisphere is an adjacent section stained for GluR-1. The dark area on the GluR-1 stained hemisphere is the DNH, an area used as a landmark to locate Cluster N. The same area around the DNH on the Zenk stained hemisphere shows considerable expression. Scale bar is 1500 µm.

No significant difference in the dorsomedial hippocampus between experimental groups was observed in a two-way ANOVA (Fig 7; $F_{(2,19)} = 0.54$; $p = 0.59$). A significant difference was observed between anterior and posterior portion of the dorsomedial hippocampus ($F_{(1, 19)} = 28.3$; $p < 0.0001$). No significant interaction between experimental group and neuroanatomical orientation was observed $F_{(2, 19)} = 0.69$; $p = 0.51$. No significant differences in the dorsolateral hippocampus between experimental groups was observed in a two-way ANOVA (Fig 7; $F_{(2,19)} = 1.77$; $p = 0.20$). A significant difference was observed between anterior and posterior parts of the dorsolateral hippocampus ($F_{(1, 19)} = 27.98$; $p < 0.0001$). No significant interaction between experimental group and neuroanatomical orientation was observed $F_{(2, 19)} = 0.39$; $p = 0.68$. No significant differences in the ventral hippocampus between experimental groups were observed in a two-way ANOVA (Fig 7; $F_{(2,19)} = 1.12$; $p = 0.33$). No significant difference was observed between the anterior and posterior subregions ($F_{(1, 19)} = 4.18$; $p = 0.06$). No significant interaction between experimental group and neuroanatomical orientation was observed $F_{(2, 19)} = 1.78$; $p = 0.12$.

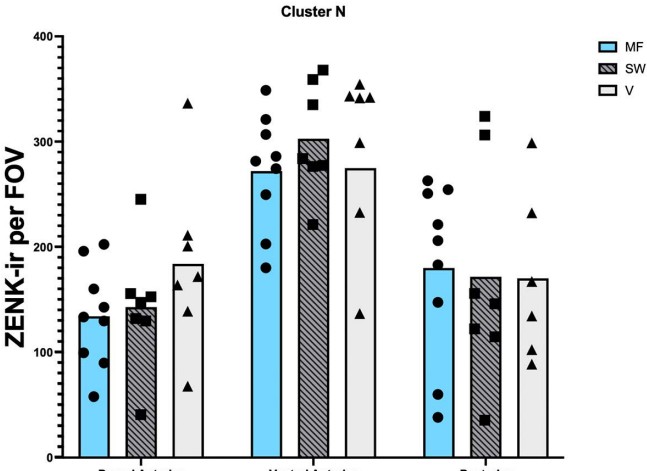

**Fig 6. Zenk immunoreactivity per field of view (FOV) by experimental group and brain subregion in Cluster N.** No differences were found between the experimental groups, but there was an effect of brain subregion, with more Zenk expression found in the ventral anterior Cluster N subregion compared to the other two. Each data point denotes the average cell count of one bird.

**Table 1. Interhemispheric comparisons of cell counts in Cluster N subregions.**

| Subregion | t(df) | p-value | Left Hemisphere (Mean±SEM) | Right Hemisphere (Mean±SEM) |
|---|---|---|---|---|
| Dorsal Anterior | t20=0.7426 | 0.47 | 148.1±63.49 | 143.8±65.35 |
| Ventral Anterior | t20=1.30 | 0.21 | 294.00±12.64 | 277.30±17.47 |
| Posterior dorsal | t19=1.73 | 0.10 | 156.30±17.97 | 167.70±18.17 |

*Note.* Two brains were excluded from interhemispheric comparisons in all subregions due to ambiguous hemisphere identification during histology. An additional bird was excluded in the posterior dorsal subregion as the tissue was damaged, and no cell counts were obtained for that particular subregion. No significant interhemispheric differences were observed; thus, hemispheric values were averaged for subsequent analyses.

**Table 2. Interhemispheric comparisons of cell counts in Hippocampal subregions.**

| Subregion | t(19) | p-value | Left Hemisphere (Mean±SEM) | Right Hemisphere (Mean±SEM) |
|---|---|---|---|---|
| Anterior dorsomedial | 0.5002 | 0.6227 | 58.25±8.09 | 62.17±11.00 |
| Posterior dorsomedial | 0.9596 | 0.3493 | 33.55±5.58 | 32.63±6.04 |
| Anterior dorsolateral | 0.1081 | 0.9150 | 183.8±19.11 | 184.8±21.49 |
| Posterior dorsolateral | 0.8775 | 0.3912 | 133.77±17.95 | 128.1±18.70 |
| Anterior ventral | 1.0570 | 0.3036 | 45.02±5.48 | 46.83±6.04 |
| Posterior ventral | 0.6466 | 0.5256 | 38.87±5.76 | 39.18±5.69 |

*Note*: Two brains were excluded from interhemispheric comparisons due to ambiguous hemisphere identification during histology. No significant interhemispheric differences were observed; thus, hemispheric values were averaged for subsequent analyses.

## Discussion

Previous studies have drawn the conclusion that Cluster N, a pattern of neural activation in the anatomical region of the visual Wulst, is specific to nocturnally migrating songbirds and that it is vital for processing magnetic compass information [1,15]. In this study, our main objective was to ask if a non-migratory bird, the zebra finch, also shows Cluster N activation.

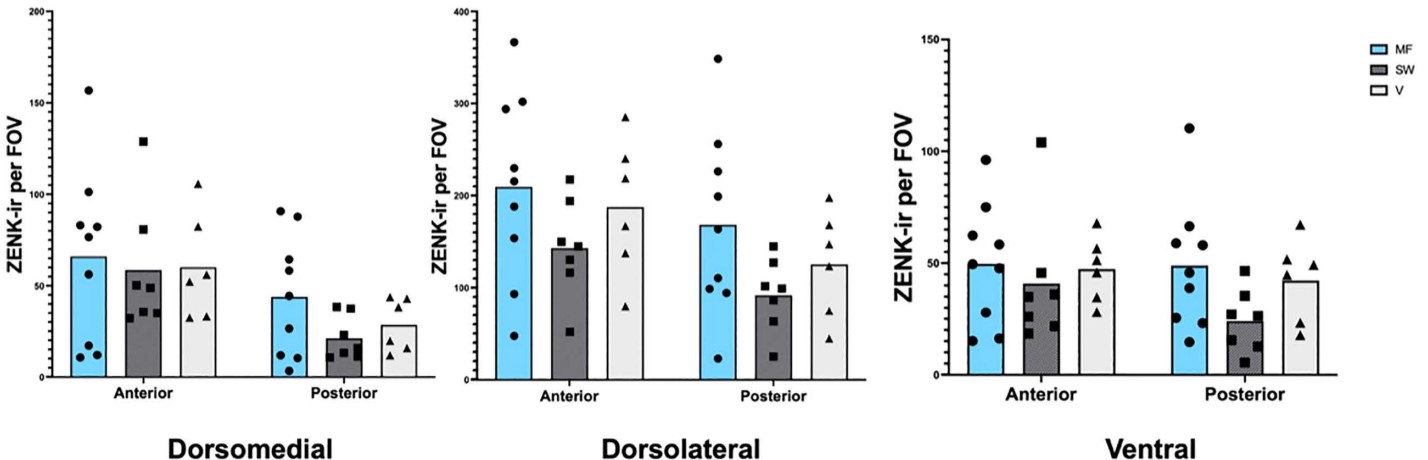

**Fig 7. Zenk immunoreactivity per FOV by experimental group and neuroanatomical orientation in the hippocampus.** Blue bars denote the static magnetic field group, hatched bars denote the sweeping magnetic field group, and light grey bars denote the vertical magnetic field group. No differences were found between the experimental groups, but there was an effect of neuroanatomical orientation in two of the subregions, with more Zenk expression found in the anterior dorsomedial hippocampus compared to the posterior dorsomedial hippocampus, and more Zenk expression found in the anterior dorsolateral hippocampus compared to the posterior dorsolateral hippocampus. Each data point denotes an individual average cell count per bird.

That is, we asked if the region of the visual Wulst that is active at night in migratory songbirds is also active in zebra finches when using magnetic compass cues to orient. We hypothesized that if Cluster N activity reflects the processing of magnetic compass information, zebra finches solving a spatial orientation task using their magnetic compass would show Cluster N activation similar to that observed in migratory species, given strong evidence that they use a light-dependent magnetic compass.

We did observe Zenk immunoreactivity in the visual Wulst region of zebra finches (Fig 5), defined as Cluster N in other species. While we did not observe differences in Zenk activation between the experimental groups, it is noteworthy that Zenk immunoreactivity in Cluster N was clearly visible in birds of all three groups (Fig 6). Thus, our results demonstrate the presence of Cluster N like immunoreactivity in a non-migratory songbird, but do not provide evidence for a condition-dependent modulation of this immunoreactivity.

Unlike prior empirical studies on Cluster N employing a sagittal plane perspective, we used a coronal plane to ensure optimal hippocampal tissue visualization. Although this is different from other studies that generally use sagittal sections, our results offer compelling support for the idea that Cluster N is not necessarily exclusive to migratory songbirds, with a few caveats. We also observed a notable difference between the different sampling regions of Cluster N. Cluster N extends across a substantial expanse of the hyperpallium and mesopallium. In the original discovery of Cluster N, Mouritsen et al. [1] reported that Cluster N takes up ~40% of the hyperpallium and dorsal mesopallium. This large size of Cluster N poses a challenge for researchers using immunohistochemistry, particularly when seeking to discern differences in activity, as activation varies across regions of Cluster N [27].

Our results align with findings from previous studies describing Cluster N in non-migratory songbirds, including zebra finches (2, 14, but see 1). Keary & Bischof [14] found higher levels of c-Fos in an anterior sampling region of the HD, which is anatomically close to our ventral anterior Cluster N sampling window. Our sampling window in the HD also had the highest amount of Zenk expression compared to our two other sampling regions in Cluster N. It is possible, therefore, that Keary & Bischof [14] were capturing Cluster N activity. However, an important difference between our findings and those of Keary & Bischof is that they reported significant hippocampal and hyperpallial (HD) activation in zebra finches

passively exposed to a continuously rotating magnetic field, whereas we observed no differences in Zenk expression between conditions during an active spatial orientation task. Several factors may account for this discrepancy. First, the magnetic stimulus used by Keary & Bischof involved strong, continuous 180° oscillations, which may be a more salient or disruptive cue than the static, sweeping, or vertically oriented magnetic fields we used here. Second, passive perception of magnetic field changes may engage neural circuits differently than active navigation. Third, Keary & Bischof used c-Fos, whereas we used Zenk, and differences in these markers may contribute to variation in sensitivity to experimental manipulations. Finally, Keary & Bischof analysed a broader set of brain regions, whereas our analyses focused specifically on Cluster N and the hippocampus. While Keary & Bischof detected condition-dependent effects without a sham control, the absence of a true negative control in our current study represents a limitation that may have reduced our ability to detect subtle differences in activation and should be considered when interpreting our results.

Contrary to our expectations, we did not detect any differences in Zenk expression within Cluster N or the hippocampus between the static magnetic field group, the sweeping magnetic field group, and the vertical magnetic field group. Our results confirm previous findings that exposure to changing and zero magnetic fields does not lead to changes in Cluster N activity. Garden warblers (*Sylvia borin*) exposed to a natural (static) magnetic field, a zero magnetic field or a magnetic field changing direction by 120° every 5 min did not show any differences in activation of Cluster N between groups [2]. Zebra finches exposed to a magnetic field changing direction by 120° per sec showed a slight, but not significant, increase in the anterior region of the HD where Cluster N is located compared to birds exposed to the natural magnetic field [14]. Liedvogel et al. [2] argued that the sampling of magnetic field information should happen constantly in the presence of light, irrespective of the properties of the magnetic field. According to the radical-pair mechanism, the magnetic field modulates the light-induced radical pairs by changing the interconversion between the singlet and triplet excited states, resulting in different ratios between the two states which changes the downstream signal [10–12]. Irrespective of whether an external magnetic field is present or not, static or changing, the brain will process the signals from the different magnetoreceptors in the retina to form a magnetic modulation pattern which informs the birds about the three-dimensional alignment of the magnetic field [2,12]. It is therefore reasonable to assume that birds in all groups may have been processing magnetic compass information even if they were not able to use it to locate food. Still, it is difficult to explain why we did not find a difference in activation between the sweeping magnetic field that constantly changed direction and the static or vertical magnetic field conditions. It is possible that the earth-strength magnetic field stimuli used in all these studies were too weak to elicit detectable changes in Zenk expression and/or that the method is not sensitive enough to detect such changes [2]. Based on our results, we cannot draw any conclusions as to whether Cluster N is involved in the processing of magnetic compass information in our zebra finches.

Previous research has suggested that migratory restlessness is positively correlated with activation of Cluster N. Migratory white-throated sparrows (*Zonotrichia albicollis*) exhibiting migratory restlessness during the night showed an increase in activity of Cluster N compared to birds tested during the migration season, but sitting still in a cage in dim light at night prior to being euthanized [8]. Other studies looking specifically at the influence of migratory restlessness on the activation of Cluster N, however, found activation in Cluster N also in birds sitting still in a cage at night [3,6,9]. Indeed, in the majority of published studies reporting Cluster N activity, the birds were sitting still, but awake, in a dimly lit cage (e.g., 1, 2, 5, 7). All of the zebra finches actively searched the maze during the behavioural experiment, and we had no control group sitting still in the arena or a cage. To assess whether variation in Cluster N activation might be attributed to differences in general activity across treatment groups, we conducted an analysis of movement during the orientation task. We found significant differences in activity across treatment groups, with birds in the magnetic field group being the most active and those in the vertical magnetic field group being the least active (Fig 4). This was also reassuring, considering that in the behavioural orientation analyses, birds in the MF group did not significantly orient in the correct direction (Fig 3), and there was no correlation between orientation performance and Zenk expression in Cluster N. The finding that the MF birds moved more than the birds in the other groups, does demonstrate some behavioural response, in that they were likely

searching more actively. In supplementary analyses, we found that activity levels did not correlate with Zenk immunoreactivity in Cluster N in any group except one subregion (anterior dorsal), in one group (vertical MF). These supplementary analyses align with our findings here, that we found no differences in Zenk immunoreactivity between the three groups. These results suggest that general movement alone is unlikely to account for Cluster N activation observed in our study.

The intensity of light appears to have the largest effect on the activation of Cluster N, with no or little activation after exposure to darkness or bright (day)light and pronounced activation after exposure to dim (moon)light (e.g., 1–5, 7, 9). We tested our zebra finches under green light of an intensity of 27 mW/m$^2$, which is brighter than the dim-light conditions used in other studies (~1 mW/m$^2$), but much dimmer than the bright-light conditions resulting in no or only little activation of Cluster N (>200 mW/m$^2$; cf. 5, 7). Further experiments testing zebra finches under much brighter light or in darkness would be needed to clarify this issue.

To contextualize these findings, we include a supplemental summary table (see Supplemental table 1), which synthesizes Cluster N activation outcomes across all other known studies of Cluster N. This table includes both published studies and some unpublished results from personal communications. The table includes study species, light, activity, migratory restlessness, magnetic field (whether it was natural or manipulated), Zenk or c-Fos in Cluster N, the sample size, and some additional comments. This summary supports the conclusion that light intensity seems to be the most consistent predictor of Cluster N activity. While there are a few exceptions, studies reporting Cluster N activation generally use dim light, regardless of other factors. Our results here, with the non-migratory zebra finch, align well with this pattern.

In addition to the lack of group differences in Cluster N, we did not find any differences in activity in the hippocampus between the three magnetic field groups. As mentioned above, birds in all groups could have been processing magnetic compass information, potentially causing activation of the hippocampus in all groups. It's likely that birds in all groups were using their hippocampus to try to solve the spatial orientation task, regardless of whether they had access to useful magnetic compass information or not. It is feasible to assume that birds that had no useful magnetic compass information available in their search reverted to other strategies, like random search or the use of topographic landmarks in the arena, indicated by the tendency of the vertical and the sweeping magnetic field group to orient topographically towards north (Fig 3). With either strategy, the birds may still use the hippocampus in the same way as if they were searching the arena with a magnetic compass.

In conclusion, we have shown that the pattern of activation in the hyperpallium and mesopallium of songbirds, described as Cluster N, is present in zebra finches during a spatial orientation task. However, we did not find evidence that Cluster N activation is modulated by magnetic field conditions in the present study. It is unclear whether the activation of Cluster N was induced by the use of magnetic compass information when the zebra finches explored the maze, whether it was induced by the dim light environment, or an interaction of the two. Nevertheless, the presence of Cluster N like immunoreactivity in non-migratory songbirds raises new intriguing questions on the function of Cluster N and the location of the processing of geomagnetic information in the brains of birds.

## Supporting information

**S1 Table. Summary of published studies examining immediate early gene expression in Cluster N in songbirds.** Included is information on light conditions, the birds' activity, presence of migratory restlessness and magnetic field conditions during exposure before sacrifice, where available. ZENK/c-Fos expression in cluster N is categorized into 0 (no activation), + (little activation), ++ (moderate activation), and +++ (high activation). (DOCX)

**S1 File. Zebra Finch Movement Analyses.** Movement analyses comparing Zenk expression in each Cluster N subregion, and birds' movement. Analyses of each subregion is also accompanied by a scatter plot with Zenk expression on the

 

y-axis (Zenk immunoreactivity per field of view), and birds' movement on the x-axis (the total activity of the bird in the 30 minute period, measured in pixels).
(DOCX)

## Acknowledgments

Thank you to all members at the Advanced Facility for Avian Research for feedback and support on this project. Thank you to David F. Sherry, who contributed advice during the early stages of this project. We would also like to thank reviewers for helpful comments on the manuscript. Thank you to Leonardo Cui, Calista Henry, Bram Richmond, Breanna Yun, and Jiexi Wu for assistance with microscope photomicrographs and general lab assistance.

## Author contributions

**Conceptualization:** Madeleine IR Brodbeck, Rachel Muheim, Scott A MacDougall-Shackleton.

**Data curation:** Madeleine IR Brodbeck.

**Formal analysis:** Madeleine IR Brodbeck.

**Funding acquisition:** Rachel Muheim, Scott A MacDougall-Shackleton.

**Investigation:** Madeleine IR Brodbeck, Rachel Muheim, Atticus Pinzon-Rodriguez, Scott A MacDougall-Shackleton.

**Methodology:** Madeleine IR Brodbeck.

**Project administration:** Rachel Muheim, Scott A MacDougall-Shackleton.

**Resources:** Rachel Muheim, Scott A MacDougall-Shackleton.

**Visualization:** Madeleine IR Brodbeck.

**Writing – original draft:** Madeleine IR Brodbeck.

**Writing – review & editing:** Madeleine IR Brodbeck, Rachel Muheim, Atticus Pinzon-Rodriguez, Scott A MacDougall-Shackleton.

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
