## [Decision Letter · Decision Letter 0]

17 Sep 2025

Dear Dr. Brodbeck,

Thank you for submitting your manuscript to PLOS ONE. After careful consideration, we feel that it has merit but does not fully meet PLOS ONE’s publication criteria as it currently stands. Therefore, we invite you to submit a revised version of the manuscript that addresses the points raised during the review process.

The reviewers were positive about the importance of your research and the findings, but there are substantive issues that each reviewer raised that must be addressed before we can consider the manuscript for publication. A satisfactory response to all of the reviewers' concerns is required for resubmission.  Please note two particular issues issues raised, first is the absnece of a negative control (Reviewer 2) and second is the suggested re-analysis focusing on individual differences given the heterogeneity of your results (Reviewer 1). I suggest running a negative control to strengthen your findings of your null results or providing a clear and detailed explanation why a negative control, such as that proposed by the reviewer, is beyond the scope of this publication. In addition, exploring the individual differences is essential to confirm the null results.

We look forward to receiving your revised manuscript.

Kind regards,

Brenton G. Cooper, Ph.D.

Academic Editor

PLOS ONE

[This research was supported by Natural Sciences and Engineering and Research Council of Canada (https://www.nserc-crsng.gc.ca/index_eng.asp) Discovery Grants awarded to SAM-S 21738, and by grant 2019-03620 from the Swedish Research Council (Vetenskapsrådet; https://www.vr.se/english.html) to RM. MIRB was supported by a Natural Sciences and Engineering Research Council of Canada Doctoral Scholarship.].

Additional Editor Comments:

Reviewer #1:

Reviewer #2:

Reviewers' comments:

Reviewer's Responses to Questions

**Comments to the Author**

1. Is the manuscript technically sound, and do the data support the conclusions?

Reviewer #1: Yes

Reviewer #2: Yes

2. Has the statistical analysis been performed appropriately and rigorously?

Reviewer #1: Yes

Reviewer #2: Yes

3. Have the authors made all data underlying the findings in their manuscript fully available?

Reviewer #1: Yes

Reviewer #2: Yes

4. Is the manuscript presented in an intelligible fashion and written in standard English?

Reviewer #1: Yes

Reviewer #2: Yes

Reviewer #1: The manuscript titled “Brain activity in Cluster N and the hippocampus in non-migratory zebra finches completing a spatial orientation task using magnetic compass information” addresses an interesting and relevant topic. However, I believe that substantial revisions and re-analysis of the data are required before it can be considered for publication.

Major points

1) Behavioral procedure: The description of the behavioral procedure is incomplete. It is unclear whether the animals received adaptation trials before training or whether food deprivation was applied. Performance data during the training phase should be provided to clarify the baseline conditions and ensure the reliability of the results.

2) Individual-based analysis: Both behavioral data and Zenk expression results show considerable individual differences, indicating a heterogeneous sample. An individual-based analysis is therefore necessary. The authors should classify subjects according to performance (e.g., good vs. poor performers) and then analyze Zenk-positive cells within these subgroups. This would allow a more accurate interpretation of the data and may reveal patterns obscured by averaging across all individuals.

Reviewer #2: This manuscript explores an interesting and timely question: whether Cluster N is active in a non-migratory songbird during magnetic compass use. The study is well conducted, and combining behavioural testing with immediate early gene expression is a valuable approach. The results are largely negative, with no significant differences in orientation or Zenk expression observed between magnetic conditions. Such negative findings are entirely suitable for PLOS ONE, provided the conclusions are appropriately cautious and the limitations are clearly acknowledged. I suggest publication after major revision, provided the authors address the following points to better align their conclusions with the data and acknowledge the study’s key limitations.

Major comments:

In this study, zebra finches were trained to locate food in a plus maze using magnetic compass cues and then tested under static, sweeping, or vertical magnetic fields. While orientation was not significant, activity levels differed across conditions. These behavioural results are important but currently underemphasised. The differences in activity (Supplementary Figure 1) show that the birds perceived the different magnetic stimulations. This figure should be moved into the main Results section and incorporated into the Discussion. Emphasising these behavioural differences would strengthen the manuscript by highlighting that the birds responded to the magnetic manipulations, even though orientation itself was not statistically significant.

Another concern is that all three experimental conditions involved an active magnetic field. A true negative control (e.g., coil off, sham, or near-zero magnetic field) is absent. Such a condition would be essential to demonstrate whether Cluster N activation depends on magnetic input rather than dim light or task context. Please explicitly acknowledge this limitation in the Discussion.

Zenk expression did not differ between magnetic conditions, and without a dim-light or no-stimulus control it cannot be determined whether the observed signal reflects specific activation of Cluster N or merely baseline Zenk expression. It is valid to note that Cluster N is anatomically present in zebra finches, but functional activation under the test conditions cannot be concluded. Nevertheless, the manuscript sometimes states that Cluster N is “active” which overstates the findings. The conclusions should be tempered, and the Discussion should explicitly acknowledge the possibility that the observed Zenk levels reflect baseline activity rather than magnetically or light-induced activation.

Furthermore, the difference with Keary & Bischof (2012), which is used as a comparison, warrants a fuller discussion. In their study, zebra finches passively exposed to a strongly rotating magnetic field showed significant hippocampal activation (c-Fos) and a trend in the hyperpallium, whereas the present study found no differences between conditions (Zenk) during active navigation. Several factors could explain this discrepancy: (i) stimulus salience (continuous 180° oscillations versus static, sweeping, or vertical fields); (ii) passive perception versus active navigation; (iii) use of different immediate early gene markers (c-Fos versus Zenk); and (iv) the broader set of brain regions analysed by Keary & Bischof compared to the focus here on Cluster N and hippocampus. Notably, while Keary & Bischof still detected differences without a sham condition, the absence of a true negative control is a more serious limitation in this study and should be explicitly acknowledged.

Minor comments:

In the Introduction, please clarify whether Cluster N is primarily a visual area, a magnetic-processing area, or both. Currently, it seems that evidence for magnetic compass processing is limited, while light dependence is more firmly established. A clearer framing would help set up the rationale for your study.

Please use standard nomenclature throughout the manuscript: write zenk and c-fos in italics when referring to the genes, and use Zenk and c-Fos (not italicized, first letter capitalised) when referring to the proteins (for c-Fos you may also simply write Fos). Avoid all-caps "ZENK," which is uncommon in avian literature and mainly used for human gene symbols. For example, on page 17 the sentence “…Keary & Bischof found higher levels of c-fos…” should read “c-Fos.”

Please also clarify figure legends (e.g., explain orientation arrows for readers unfamiliar with orientation analyses).

Report exact p-values (and effect sizes where possible) rather than describing non-significant results as “trends.”

Ensure consistent terminology for brain regions (e.g., always “Cluster N”; specify whether “hippocampus” refers to the whole formation or a subdivision).

Light intensity (green LED, 27 mW/m²) is unusually high compared to prior Cluster N studies. Please elaborate on this in Methods/Discussion, as it may affect Cluster N activation.

I hope these comments are helpful in revising the manuscript.

.

Reviewer #1: **Yes:** Shigeru WatanabeShigeru WatanabeShigeru WatanabeShigeru Watanabe

Reviewer #2: **Yes:** Uwe MayerUwe MayerUwe MayerUwe Mayer

---

## [Author Response · Author response to Decision Letter 1]

17 Mar 2026

Dear Dr. Cooper,

We thank you and the reviewers for your thoughtful and constructive feedback on our manuscript, “Brain activity in Cluster N and the hippocampus in non-migratory zebra finches completing a spatial orientation task using magnetic compass information.”

We have revised the manuscript in response to the comments. Specifically, we have moved Figure 1 from the supplementary to the main manuscript, and we have expanded the discussion in several ways. Unfortunately, it is not possible for us to run more birds through this experiment, as the host lab that ran these studies is now closed, but we have included the lack of a negative control in the discussion.

We believe these changes have strengthened the manuscript, thank the reviewers in the acknowledgements, and we address each comment directly below.

Sincerely,

Madeleine IR Brodbeck

Response To Reviewers

Reviewer #1: The manuscript titled “Brain activity in Cluster N and the hippocampus in non-migratory zebra finches completing a spatial orientation task using magnetic compass information” addresses an interesting and relevant topic. However, I believe that substantial revisions and re-analysis of the data are required before it can be considered for publication.

MB: Thank you for your comments.

Major points

1) Behavioral procedure: The description of the behavioral procedure is incomplete. It is unclear whether the animals received adaptation trials before training or whether food deprivation was applied. Performance data during the training phase should be provided to clarify the baseline conditions and ensure the reliability of the results.

MB: We have now added additional detail on the training procedure in the Behavioural Procedure.

2) Individual-based analysis: Both behavioral data and Zenk expression results show considerable individual differences, indicating a heterogeneous sample. An individual-based analysis is therefore necessary. The authors should classify subjects according to performance (e.g., good vs. poor performers) and then analyze Zenk-positive cells within these subgroups. This would allow a more accurate interpretation of the data and may reveal patterns obscured by averaging across all individuals.

MB: Thank you for this comment; we had a similar thought. An analysis we originally did, began with splitting the birds into “well-oriented” and “poor-oriented”, to see if we could find any difference across the three regions we sampled of Cluster N. We ran a 2-way ANOVA, with orientation performance, and brain subregion as our factors. We found no significant effect of orientation performance. However, we found the binarization of well oriented vs poor oriented to be a bit arbitrary, so we instead looked at a correlation of orientation performance and immunoreactive cells in Cluster N. This is indeed included in the manuscript, see section “ZENK Immunoreactivity” under “Methods” for descriptions of our analysis, and the “Cluster N” section under “Results”, for our results. Ultimately, we found no correlation between success on the task and immunoreactivity in Cluster N. To make this analysis and result more apparent, we have now added a line in the discussion about this result.

Reviewer #2: This manuscript explores an interesting and timely question: whether Cluster N is active in a non-migratory songbird during magnetic compass use. The study is well conducted, and combining behavioural testing with immediate early gene expression is a valuable approach. The results are largely negative, with no significant differences in orientation or Zenk expression observed between magnetic conditions. Such negative findings are entirely suitable for PLOS ONE, provided the conclusions are appropriately cautious and the limitations are clearly acknowledged. I suggest publication after major revision, provided the authors address the following points to better align their conclusions with the data and acknowledge the study’s key limitations.

Major comments:

In this study, zebra finches were trained to locate food in a plus maze using magnetic compass cues and then tested under static, sweeping, or vertical magnetic fields. While orientation was not significant, activity levels differed across conditions. These behavioural results are important but currently underemphasised. The differences in activity (Supplementary Figure 1) show that the birds perceived the different magnetic stimulations. This figure should be moved into the main Results section and incorporated into the Discussion.

MB: We have now added Supplementary Figure 1 into the manuscript. It is now Figure 3, and subsequent figures have now also been renamed. We have now added a brief section on the analysis into the Methods, and similarly to the Results. We have also added our interpretation of this into the discussion.

Emphasising these behavioural differences would strengthen the manuscript by highlighting that the birds responded to the magnetic manipulations, even though orientation itself was not statistically significant.

MB: Thank you for your suggestion, we agree and have now added further discussion.

Another concern is that all three experimental conditions involved an active magnetic field. A true negative control (e.g., coil off, sham, or near-zero magnetic field) is absent. Such a condition would be essential to demonstrate whether Cluster N activation depends on magnetic input rather than dim light or task context. Please explicitly acknowledge this limitation in the Discussion.

MB: Indeed. We have now included a discussion of this limitation in our discussion.

Zenk expression did not differ between magnetic conditions, and without a dim-light or no-stimulus control it cannot be determined whether the observed signal reflects specific activation of Cluster N or merely baseline Zenk expression. It is valid to note that Cluster N is anatomically present in zebra finches, but functional activation under the test conditions cannot be concluded. Nevertheless, the manuscript sometimes states that Cluster N is

“active” which overstates the findings.

MB: We have amended the wording of “active”. We have also clarified Cluster N as a pattern of activation. Structurally, all songbirds have this region, but Cluster N is a pattern of activation in the hyperpallium and mesopallium.

The conclusions should be tempered, and the Discussion should explicitly acknowledge the possibility that the observed Zenk levels reflect baseline activity rather than magnetically or light-induced activation.

MB: Thank you; we believe the conclusions have been sufficiently tempered, especially with including the information in your next section.

Furthermore, the difference with Keary & Bischof (2012), which is used as a comparison, warrants a fuller discussion. In their study, zebra finches passively exposed to a strongly rotating magnetic field showed significant hippocampal activation (c-Fos) and a trend in the hyperpallium, whereas the present study found no differences between conditions (Zenk) during active navigation. Several factors could explain this discrepancy: (i) stimulus salience (continuous 180° oscillations versus static, sweeping, or vertical fields); (ii) passive perception versus active navigation; (iii) use of different immediate early gene markers (c-Fos versus Zenk); and (iv) the broader set of brain regions analysed by Keary & Bischof compared to the focus here on Cluster N and hippocampus. Notably, while Keary & Bischof still detected differences without a sham condition, the absence of a true negative control is a more serious limitation in this study and should be explicitly acknowledged.

MB: Thank you for this interpretation. We have included much of this information in our discussion of Keary and Bischof, as well as pointing to our lack of a negative control group.

Minor comments:

In the Introduction, please clarify whether Cluster N is primarily a visual area, a magnetic-processing area, or both. Currently, it seems that evidence for magnetic compass processing is limited, while light dependence is more firmly established. A clearer framing would help set up the rationale for your study.

MB: An excellent point. We do not know if Cluster N is primarily a visual area, a magnetic processing area, or both. Indeed, the evidence for magnetic compass processing is limited, and there is more evidence that it is visual. However, there is a body of work that places Cluster N as a magnetic region, which is what we hoped to communicate in our introduction. We believe that we have made this clear in our opening paragraph, but more specific feedback on how to make this clearer is welcome.

Please use standard nomenclature throughout the manuscript: write zenk and c-fos in italics when referring to the genes, and use Zenk and c-Fos (not italicized, first letter capitalised) when referring to the proteins (for c-Fos you may also simply write Fos). Avoid all-caps "ZENK," which is uncommon in avian literature and mainly used for human gene symbols. For example, on page 17 the sentence “…Keary & Bischof found higher levels of c-fos…” should read “c-Fos.”

MB: Acknowledged, and fixed.

Please also clarify figure legends (e.g., explain orientation arrows for readers unfamiliar with orientation analyses).

MB: In Figure 1 and 2, an additional sentence has been added about the anatomical orientation “compass”. For Figure 3, we have adjusted the wording slightly to clarify the meaning of the arrows.

Report exact p-values (and effect sizes where possible) rather than describing non-significant results as “trends.”

MB: We have removed any instance of the word “trend”.

Ensure consistent terminology for brain regions (e.g., always “Cluster N”; specify whether “hippocampus” refers to the whole formation or a subdivision).

MB: Thank you, any instances of “cluster N” have now been changed to “Cluster N”. Most references to the hippocampus refer to the whole structure. We have clarified the use of “hippocampus” in a few instances in the revised manuscript.

Light intensity (green LED, 27 mW/m²) is unusually high compared to prior Cluster N studies. Please elaborate on this in Methods/Discussion, as it may affect Cluster N activation.

MB: We have added additional information on the green light in manuscript on the light intensity, under “experimental set up” on page 6. We also discuss light intensity on page 19 in the discussion.

I hope these comments are helpful in revising the manuscript.

MB: Thank you very much for your feedback, the comments have indeed been helpful.

---

## [Decision Letter · Decision Letter 1]

12 Apr 2026

Brain activity in Cluster N and the hippocampus in non-migratory zebra finches completing a spatial orientation task using magnetic compass information

PONE-D-25-37676R1

Dear Dr. Brodbeck,

We’re pleased to inform you that your manuscript has been judged scientifically suitable for publication and will be formally accepted for publication once it meets all outstanding technical requirements.

Kind regards,

Brenton G. Cooper, Ph.D.

Academic Editor

PLOS One

Additional Editor Comments (optional):

Reviewers' comments:

Reviewer's Responses to Questions

**Comments to the Author**

Reviewer #1: All comments have been addressed

Reviewer #2: All comments have been addressed

2. Is the manuscript technically sound, and do the data support the conclusions?

Reviewer #1: Yes

Reviewer #2: Yes

3. Has the statistical analysis been performed appropriately and rigorously?

Reviewer #1: Yes

Reviewer #2: Yes

4. Have the authors made all data underlying the findings in their manuscript fully available?

Reviewer #1: Yes

Reviewer #2: Yes

5. Is the manuscript presented in an intelligible fashion and written in standard English?

Reviewer #1: Yes

Reviewer #2: Yes

Reviewer #1: I have read the revised MS and correspondence between the authors and reviewers. The MS has been well revised and I can recommend it for publication.

Reviewer #2: I am happy with the revised version of the manuscript.

PS: While reading, I noticed a small typo: on page 10, “DNH = dorsolaternal…” should be written without the “n”.

.

Reviewer #1: No

Reviewer #2: **Yes:** Uwe MayerUwe MayerUwe MayerUwe Mayer

---

## [Editor Report · Acceptance letter]

PONE-D-25-37676R1

PLOS One

Dear Dr. Brodbeck,

I'm pleased to inform you that your manuscript has been deemed suitable for publication in PLOS One. Congratulations! Your manuscript is now being handed over to our production team.

Kind regards,

on behalf of

Dr. Brenton G. Cooper

Academic Editor

PLOS One